# Semantic Proximity for Redundancy-Aware Context Compression in Large Language Models

## Abstract

LLMs are increasingly bottlenecked by fixed context windows, motivating principled compression of conversational histories. We study semantic-redundancy–aware compression, in which we pair human–assistant turns, embed them, and summarize those that are most semantically overlapping. We introduce STAE (Semantic-Temporal Aware Eviction), a centroid–temporal hybrid policy that scores each pair by a convex combination of semantic distance to a conversation centroid and recency (weighted by $\beta$), alongside an inverted variant and a cluster-aware compressor that summarizes whole embedding-space clusters. Crucially, redundancy is detected from embeddings using lightweight centroid/cluster arithmetic without extra LLM calls, reducing token usage and inference cost. To evaluate retrieval under compression, we augment LongMemEval with a 20-needle-per-dialogue benchmark, addressing the brittleness of single-needle tests and enabling finer-grained measurement of information retention. On this benchmark, summarizing pairs closest to the centroid outperforms FIFO across compression regimes, while compression of those furthest from the centroid degrades at stricter budgets; moreover, local STAE within temporal or semantic groups closely matches a strong temporal upper bound and consistently surpasses global eviction at the same ECR, with inverted (evict-lowest) preserving more needles. We also show that clustered summarization of semantically or temporally similar message pairs provides a strong chunking strategy for compression. The takeaway is simple and actionable: compress where redundancy is highest, measured explicitly via semantic similarity in embedding space, while freeing tokens with minimal loss.

## 1 Introduction

Large Language Models (LLMs) operate under fixed input windows, forcing practitioners to either truncate long histories or compress them aggressively. In interactive, multi-session settings, naively shrinking context often discards task-relevant facts and undermines long-horizon reasoning. The core problem is to selectively compress conversational history so that downstream question answering retains as many factual "needles" as possible under strict token budgets.

At the same time, production systems increasingly rely on long-term, multi-topic memory (teaching assistants, coding copilots, agents with episodic logs). Simply buying larger windows is costly and empirically insufficient: information placed in the middle is often lost or misused, and FIFO-style eviction remains content-agnostic. A practical, LLM-agnostic, token-efficient compressor that preserves concrete facts improves reliability, cost, and latency without retraining.

Designing such a compressor requires accounting for two main characteristics of long context: (i) semantic redundancy (many turns paraphrase or elaborate the same idea) and (ii) topic drift (older segments may cease to reflect the current theme). Pure recency (FIFO) ignores content and deletes rare but crucial facts; pure relevance heuristics can overfit to local cues or be brittle to paraphrase. Moreover, budgeted summarization introduces stochastic variability and a nontrivial coupling between what is evicted and what is kept: the realized compression (tokens removed) can deviate substantially from intent, complicating evaluation.

Existing remedies only partially address these issues. Architectural solutions, such as sparse attention (Beltagy et al., 2020) and cached recurrence (Dai et al., 2019) relax scaling but do not directly optimize which tokens to keep; retrieval pipelines decouple storage but still face in-context organization. Agent memory systems such as MemGPT use FIFO queues and recursive summarization (Packer et al., 2023), remaining largely content-agnostic at eviction time. Prior content-aware methods seldom provide a tunable, unified score that trades off semantic redundancy and age, and evaluation commonly relies on single-needle tests that are brittle to cue phrasing.

To address these gaps, we propose STAE (Semantic–Temporal Aware Eviction), a lightweight policy that ranks message pairs by a convex combination of semantic redundancy, estimated from embedding-space density or a centroid-proximity proxy, and recency, then evicts highest-scoring items and summarizes the evicted content under several budgeted schemes.

## 1.1 Contributions

We summarize the key components of our contributions in the list below.

- **Augmented LongMemEval: Needle Density Benchmark.** We construct a long-dialogue evaluation with 20 atomic, source-verifiable needles per conversation, uniformly distributed across temporal segments, and measure Needle Retention Rate (NRR) under compression, reporting Effective Compression Rate (ECR) alongside accuracy for fair budget accounting (Wu et al., 2025).

- **STAE: Semantic–Temporal Aware Eviction.**
    1. *Redundancy signal $R(m)$* via kernel-density in embedding space (preferred) or a centroid-proximity proxy for speed; temporal signal $T_{\text{norm}}(m)$ via normalized age.
    2. *Unified eviction score $S(m) = \beta R + (1-\beta)T$*, with $\beta \in [0, 1]$ interpolating FIFO ($\beta=0$) and semantic-only ($\beta=1$); we also test an inverted variant (remove most atypical items first).
    3. *Global and local application.* Global over the full history, or local within (i) temporal chunks or (ii) $k$-means thematic clusters.
    4. *Handling evicted content.* (i) concatenated per-group summaries, (ii) summary-of-summaries, and (iii) fixed-budget, per-group summaries.
    5. *Evaluation protocol and ablations.* NRR–ECR curves across compression budgets and $\beta$ values, including comparisons to FIFO baselines (as in MemGPT (Packer et al., 2023)).

## 2 Related Work

### 2.1 Architectural and Retrieval-Based Approaches to Long Contexts

Self-attention scales quadratically (Vaswani et al., 2017), and yet surveys report native context windows in the $10^5$–token range (Pawar et al., 2024). Remedies include *sparse attention* (*Longformer* (Beltagy et al., 2020), *BigBird* (Zaheer et al., 2020)), recurrence with cached states (*Transformer-XL* (Dai et al., 2019)), and optimized kernels (*FlashAttention* (Dao et al., 2022)). In parallel, retrieval augmented generation (RAG) decouples storage from parametric memory (Lewis et al., 2020), with hierarchical retrieval (*RAPTOR* (Sarthi et al., 2024)) and learned when-to-retrieve (*ADAPT-LLM* (Labruna et al., 2024)). To push quality-over-quantity context, *HippoRAG* encodes experiences into a knwoledge-graph (KG) with Personalized PageRank, retrieving compact, high-yield chains that cut cost/latency while improving multi-hop QA (Jimenez Gutierrez et al., 2024). Beyond text, *RA-GRAPH* treats external graph repositories as episodic memory, retrieving a few task-relevant "toy graphs" to condition message passing and improve generalization without task-specific fine-tuning (Jiang et al., 2024). Despite these advances, NIAH tests (Laban et al., 2024) reveal a "lost in the middle" effect (Liu et al., 2023), motivating optimization of context quality and organization.

### 2.2 Intelligent In-Context Memory Management

A complementary line prioritizes what to store. The "LLMs as OS" view casts the model as a CPU over a memory hierarchy (Ge et al., 2023); *MemGPT* instantiates FIFO "RAM" and recursive sum-

marization to "disk" (Packer et al., 2023). A recent survey systematizes agent memory designs and open challenges for principled, content-aware eviction (Zhang et al., 2025). Beyond heuristics, $EM^2$ treats explicit memory updates as latent-variable inference via EM, bridging explicit (readable) and implicit (trainable) memory and outperforming static external-memory baselines on streaming inference (Yin et al., 2024); a hybrid multimodal design (*Optimus-1*) couples a Hierarchical Directed Knowledge Graph with an Abstracted Multimodal Experience Pool, yielding strong long-horizon gains and surpassing a GPT-4V baseline in Minecraft (Li et al., 2024). For long-horizon personalization, *SeCom* combines topical conversation segmentation with compressed memory units (LLMLingua-2 denoising), improving retrieval and response quality on LOCOMO and Long-MT-Bench+ (Pan et al., 2025). *EM-LLM* performs episodic segmentation with two-stage retrieval (similarity + temporal contiguity), handling practically "infinite" contexts and outperforming full-context and strong RAG/InfLLM baselines on LongBench variants at lower compute (Fountas et al., 2025). In embodied settings, *MrSteve* adds what-where-when Place Event Memory to reduce repeated failures and boost long-horizon completion in Minecraft (Park et al., 2025); from a control perspective, *Stable Hadamard Memory* introduces a differentiable erase-and-reinforce (Hadamard-gated) update that stabilizes long-horizon credit assignment and outperforms prior memory models on meta-RL and POPGym (Le et al., 2025). While influential, FIFO is content-agnostic. Evidence for content-aware policies includes relevance-based history selection (Qu et al., 2019; Xiong et al., 2021), entity memory for factual consistency (Févry et al., 2020; Zhang et al., 2022), and temporal-decay/LRU-like caches (Makhkamova & Kim, 2021). Yet a unified, tunable score combining semantic centrality and temporal decay is missing; our Semantic and Temporal Aware Eviction (STAE) addresses this and explores local application within thematic clusters.

### 2.3 EVALUATING INFORMATION PRESERVATION IN COMPRESSED CONTEXTS

Evaluation must move beyond isolated fact retrieval: NIAH alone misses synthesis and can be cue-sensitive (Goldman et al., 2024; Modarressi et al., 2025); summarization often ignores multi-facet preservation (Wang et al., 2025); and standard retrieval metrics can misalign with generated answers (Arabzadeh et al., 2024). Complementarily, *LongMemEval* probes interactive long-term memory (extraction, multi-session/temporal reasoning, knowledge updates, abstention) and articulates an indexing–retrieval–reading design with effective optimizations (Wu et al., 2025). Meanwhile, *HotpotQA* (Yang et al., 2018) and *QuALITY* (Pang et al., 2022) target static documents, and long-term open-domain dialogue raises distinct memory challenges (Xu et al., 2021). We therefore propose *Needle Density* to quantify information-preservation density under compression and evaluate our summarization and eviction strategies.

## 3 METHODOLOGY

We evaluate principled in-context memory compression strategies designed to remove redundant content while preserving question-answering fidelity. This section introduces (i) a benchmark for measuring information preservation beyond single-fact retrieval, and (ii) a semantic–temporal scoring rule that prioritizes eviction of redundant, older content.

### 3.1 BENCHMARK: NEEDLE DENSITY FOR HOLISTIC INFORMATION PRESERVATION

To evaluate beyond single-fact retrieval, we build on *LongMemEval* (Wu et al., 2025). From its large collection of real multi-turn dialogues, we select long conversations (average $\sim 100,000$ tokens) as "haystacks." Each conversation is partitioned into 20 contiguous, equal-sized segments (by message count). Using gpt-5-mini, we generate exactly one self-contained question–answer (Q&A) per segment, with atomic facts and programmatic verification that each "source-quote" is a verbatim substring of the original conversation (with retries to ensure quality). This yields $\sim 10,000$ Q&A pairs overall.

**Metric.** The Needle Retention Rate (NRR) is the percentage of a conversation's 20 questions that are answered correctly using only the compressed context produced by a given algorithm.

## 3.2 A Discrete $\varepsilon$–$\delta$ Motivation for Redundancy via Semantic Proximity

Let $\Sigma$ denote a finite vocabulary and $\Sigma^*$ the set of token sequences. Let $f : \Sigma^* \to \mathbb{R}^d$ be a sentence (or message-pair) embedding function that maps a text $x \in \Sigma^*$ to a vector $v = f(x) \in \mathbb{R}^d$. Intuitively, if two texts $x, y$ are semantically very similar, then their embeddings $f(x), f(y)$ should be close in $\mathbb{R}^d$.

**Why standard $\varepsilon$–$\delta$ continuity does not strictly apply.** Unlike real-valued inputs, $x \in \Sigma^*$ is discrete and does not admit a literal "$+\varepsilon$" perturbation. Consequently, classical continuity ("for every $\varepsilon > 0$ there exists $\delta > 0$ s.t. $\|x - y\| < \delta \Rightarrow \|f(x) - f(y)\| < \varepsilon$") is ill-posed because there is no metric over raw strings with infinitesimal neighborhoods that matches the embedding geometry.

**Latent perturbations on the natural text manifold** Despite discreteness, naturally occurring texts concentrate on a low-dimensional manifold $\mathcal{M} \subset \Sigma^*$ induced by human paraphrase operations: synonym substitutions, minor word reorderings, or local clarifications. Consider a latent operation $x \mapsto x'$ that replaces one word by a near-synonym or inserts a short parenthetical, leaving propositional content unchanged. Although $x'$ is not an infinitesimal move in $\Sigma^*$, empirically $f(x')$ is a small perturbation of $f(x)$:

$$\|f(x') - f(x)\| = \delta(x \to x') \quad \text{with} \quad \delta \text{ small whenever the edit is semantically local.}$$

Thus, while $\varepsilon$ is not defined on $\Sigma^*$, there exists a $\delta$-neighborhood in embedding space within which edits preserve meaning. In other words, there exists a radius below which $f$ maps many distinct but semantically equivalent texts to vectors that are effectively indistinguishable for downstream QA.

**Redundancy criterion (radius form).** Let $\mathbb{B}(v, r) = \{u \in \mathbb{R}^d : \|u - v\| \le r\}$ denote a closed ball. For a collection of embedded units $\{v_i\}$ (message pairs; see §3.3), we say a subset $\mathcal{C} \subseteq \{v_i\}$ is redundant at radius $r$ if $\mathrm{diam}(\mathcal{C}) \le 2r$ and all members correspond to near-paraphrases or minor elaborations of the same local content. In that case, replacing $\mathcal{C}$ by a single representative (or a short summary) is an $r$-lossy compression that preserves task answers up to the embedding-level tolerance $r$. As $r$ tends towards $0$, compression becomes less aggressive (lower loss); as $r$ increases, more vectors are consolidated (higher compression, potentially higher loss).

## 3.3 Units, Geometry, and Normalization

**Embedded units.** We operate on message pairs (user–assistant turns) to preserve conversational adjacency. Each pair $m$ is embedded as a single vector $v_m \in \mathbb{R}^d$ using `text-embedding-3-small` (1536 dims).

**Distance.** All vectors are $\ell_2$-normalized, so Euclidean distance is monotone in cosine distance: $\|\hat{v}_i - \hat{v}_j\|_2^2 = 2(1 - \cos\angle(v_i, v_j))$. We write $D(m)$ for the (normalized) distance from $m$ to a reference (e.g., a centroid).

## 3.4 Semantic–Temporal Redundancy Scoring

To prioritize removal of redundant and older content, we combine (i) semantic centrality (messages near the local theme are more likely to repeat known facts) with (ii) recency (older content is more likely to have been superseded).

Let $\mathcal{G}$ be the current grouping (global: one group; local: clusters or chunks). For a message-pair $m \in \mathcal{G}$ with embedding $v_m$, let $c_{\mathcal{G}}$ denote the (unit-norm) centroid of embeddings in $\mathcal{G}$. That is,

$$D_{\mathrm{norm}}(m) = \frac{\|\hat{v}_m - \hat{c}_{\mathcal{G}}\|_2 - \min\limits_{j \in \mathcal{G}} \|\hat{v}_j - \hat{c}_{\mathcal{G}}\|_2}{\max\limits_{j \in \mathcal{G}} \|\hat{v}_j - \hat{c}_{\mathcal{G}}\|_2 - \min\limits_{j \in \mathcal{G}} \|\hat{v}_j - \hat{c}_{\mathcal{G}}\|_2} \in [0, 1].$$

This yields a density-based redundancy score. Let $N_k(m)$ be the $k$ nearest neighbors of $m$ within group $\mathcal{G}$ (by cosine distance on unit vectors), and let $h > 0$ be a bandwidth. Define the (local) kernel density estimate

$$\rho(m) = \frac{1}{k} \sum_{j \in N_k(m)} \exp\left(-\frac{\|\hat{v}_m - \hat{v}_j\|_2^2}{2h^2}\right),$$

and its min–max normalization within $\mathcal{G}$,

$$\rho_{\mathrm{norm}}(m) \;=\; \frac{\rho(m) - \min_{j \in \mathcal{G}} \rho(j)}{\max_{j \in \mathcal{G}} \rho(j) - \min_{j \in \mathcal{G}} \rho(j)} \;\in [0, 1].$$

We define the redundancy score as

$$R(m) \;:=\; \rho_{\mathrm{norm}}(m),$$

so that high $R(m)$ means $m$ lies in a locally high-density region (many close paraphrases or minor elaborations), while low $R(m)$ flags sparse regions that are more likely to contain unique facts and should be preserved under needle-in-haystack evaluations. For efficiency, a centroid-based proxy can be used:

$$R_{\mathrm{proxy}}(m) := 1 - D_{\mathrm{norm}}(m)\,,$$

with the caveat that centrality reflects semantic typicality rather than true local density; when feasible, we prefer $\rho_{\mathrm{norm}}$.

Let $T_{\mathrm{norm}}(m) \in [0, 1]$ be the normalized age rank within $\mathcal{G}$ (1=oldest).

**Eviction score.**
$$S(m) \;=\; \beta\, R(m) \;+\; (1 - \beta)\, T_{\mathrm{norm}}(m), \qquad \beta \in [0, 1].$$

Higher $S(m)$ means more evictable: messages that are both semantically central (likely redundant) and old are evicted first. Setting $\beta = 0$ recovers a purely temporal policy; $\beta = 1$ becomes a purely semantic redundancy policy.

### 3.5 GLOBAL VS. LOCAL STRATEGIES AND SUMMARIZATION

**Global (baseline).**  Compute a single centroid over the full history and rank messages by $S(m)$ (§3.4). Evict in decreasing $S$ until the budget is met.

**Local (two-stage).**  First partition the conversation, then apply scoring and (optionally) consolidation within each partition.

**Grouping strategies.**

1. *Thematic clusters*: $k$-means on $\{\hat{v}_m\}$ with a preset $k$.
2. *Temporal chunks*: split into $k$ consecutive segments of equal size (by pairs).

**Handling evicted content.** For either grouping, we evaluate:

1. *Concatenated summaries*: summarize evicted messages per group; the final context consists of the kept messages plus the per-group summaries (with each summary budget proportional to the tokens evicted from its group).
2. *Summary of summaries*: produce per-group summaries, then compress them once more into a single meta-summary.
3. *Fixed-budget summaries*: allocate a global $B$-word budget equally across groups ($B/k$ each).

### 3.6 EFFECTIVE COMPRESSION RATE (ECR)

Because we evict discrete units (message pairs) and perform component-wise consolidations, realized compression can overshoot a target. We therefore report the Effective Compression Rate (ECR) for every NRR value:

$$\mathrm{ECR} \;=\; \frac{start\_tokens - final\_tokens}{start\_tokens},$$

where *start_tokens* is the pre-compression count and *final_tokens* is the token count after eviction and summarization.

### 3.7 EVALUATION PROTOCOL

All methods are evaluated on the Needle Density benchmark (§3.1). For each conversation, we compress the full history using a specified strategy (global or local) and scoring rule (§3.4). We then answer the 20 conversation-specific questions using only the compressed context and compute NRR. Results are presented as NRR–ECR trade-off curves across budgets, $\beta$ values, and grouping choices.

**Implementation details.** Embeddings use `text-embedding-3-small` (1536d). Vectors are $\ell_2$-normalized prior to distance computations. Centroids are computed in the normalized space and re-normalized. $k$-means is initialized with $k$-means++ and run with 10 restarts. Summaries are generated by an LLM (stochastic decoding) with a target token budget per summary but no hard cap; variability in realized summary lengths is expected and contributes to differences between intended and realized compression.

**Summary of assumptions.** (1) On naturally occurring edits (synonyms, local paraphrases), $f$ is locally stable: semantically small edits yield embedding deltas below a problem-dependent quantization margin. (2) Messages near a local centroid are more likely to restate prevously present content; hence high redundancy score $R(m)$. (3) Older messages are, ceteris paribus, less likely to reflect the current theme of very long conversations due to topic drift. The experiments quantify how these assumptions trade off against compression.

### 3.8 EXPERIMENTAL SETUP

The benchmark harness orchestrates all experiments.

- **Models.** Evicted content is summarized by `gpt-5-nano`. The final Q&A task uses `gpt-5-mini`.

- **Q&A Process.** All 20 needle questions are posed in a single prompt following the compressed context. The model is instructed to return one JSON object containing all 20 answers using `response_format={"type":"json_object"}`.

- **Parameters.** We sweep multiple compression-ratio values. STAE is tested with $\beta \in \{0, 0.5, 1.0\}$.

## 4 EXPERIMENTS

We evaluate STAE on a 20-needle long-context benchmark under both global and local (temporal/thematic) settings, varying $\beta$ and compression budgets. Results are reported as Needle Retention Rate (NRR) versus realized Effective Compression Rate (ECR) to capture the true budget–accuracy trade-off.

**Reproducibility.** Clustering and grouping were implemented using `scikit-learn` KMeans, with the number of clusters $k \in [2, 25]$ selected via silhouette analysis (it is worth noting that majority of the time 25 clusters were created). Both global centroids and temporal chunking were used as baselines. Unless otherwise noted, cluster summaries were targeted to 100 words.

### 4.1 BASELINE: PERFORMANCE OF GLOBAL STAE EVICTION

**Setup.** We evaluate global STAE by embedding every (human, assistant) pair in the full dialogue, forming a single conversation centroid, computing each pair's STAE score (centroid proximity with recency weight $\beta$), and compressing a fixed fraction of pairs. We compare two eviction policies at identical target compression ratios (CR): regular STAE (evict top-scoring pairs; dashed) and inverted STAE (evict bottom-scoring pairs; solid). Because a single summary is requested at a fixed token budget of 100 tokens, the effective compression ratio closely matches the target CR; thus we report Needle Retention Rate (NRR) only (Fig. 1).





Figure 1: **Experiment 1: Global STAE Eviction.** NRR (%) vs. target compression ratio (CR) for two eviction policies and multiple $\beta$ values. Solid lines (Lowest) consistently outperform dashed lines (Highest), especially at larger $\beta$. FIFO ($\beta$=0) shows near-symmetric behavior, indicating recency alone does not isolate unique information.

**Results.** Across all CRs, regular STAE consistently outperforms inverted STAE. The gap is most visible at stronger recency weighting ($\beta = 1.0$) and moderate CR (e.g., CR $\approx 0.5$), and persists even under aggressive compression (CR $\approx 0.7$). FIFO ($\beta = 0$) shows near-symmetric performance between regular STAE and inverted, indicating limited discrimination of semantically unique content under pure recency.

**Interpretation.** Critically, at a fixed CR we summarize the same number of pairs, and needles are temporally uniform. If STAE scoring were not about semantic similarity, swapping the eviction order (Highest $\leftrightarrow$ Lowest) should nearly invert the outcome (e.g., NRR $\approx 0.7$ vs. 0.3 for the same $\beta$), because both policies would compress indistinguishable content aside from order. We do not observe such inversion. Instead, Lowest > Highest: evicting low-score (semantically redundant) pairs preserves high-score, distinctive pairs that are more likely to contain needle information, while Highest deliberately compresses those distinctive pairs and harms NRR. The FIFO case ($\beta$=0) landing near parity reinforces this logic: without semantic discrimination, compressing "earlier vs. later" content yields essentially the same retention as the number of message-pairs to eb evicted are held constant and we have uniformly distributed needles in a temporal sense.

**Takeaway.** In global STAE, *evicting the most semantically redundant pairs* (Lowest) consistently preserves more needles than evicting the highest-scoring pairs, validating that centroid-based STAE scoring aligns with semantic similarity rather than mere recency.

## 4.2 LOCAL STAE EVICTION WITHIN GROUPS

**Setup.** We partition the dialogue into $k$ local groups and apply STAE within each group rather than globally. We study two grouping schemes: (i) temporal windows (uniform in time) and (ii) thematic clusters obtained by $k$-means in embedding space. For every group we compute a local centroid, score (human,assistant) pairs by the STAE score with recency weight $\beta \in \{0, 0.5, 1.0\}$, and evict a fraction of pairs per group. We evaluate both eviction orders: regular STAE (evict highest scores) and inverted STAE (evict lowest scores). Evicted content is handled by three summarization modes: *Concatenated Summaries*, *Summary of Summaries*, and *Fixed-Budget Summaries*. Because summarization is carried out independently in $k$=25 groups, the total number of tokens produced by the LLM deviates from the target budget; thus we report Needle Retention Rate (NRR) against the realized Effective Compression Rate (ECR).

In our benchmark, needles are uniformly distributed over time. With temporal windows and $k \geq$ #needles, each window typically contains an expected single needle. When the summarizer is instructed to preserve unique factual information per group, the needle can be retained without interference from other needles. Consequently, temporal chunking serves as a strong upper-bound baseline. Thematically clustered groups achieve very similar NRR at matched ECR, demonstrating

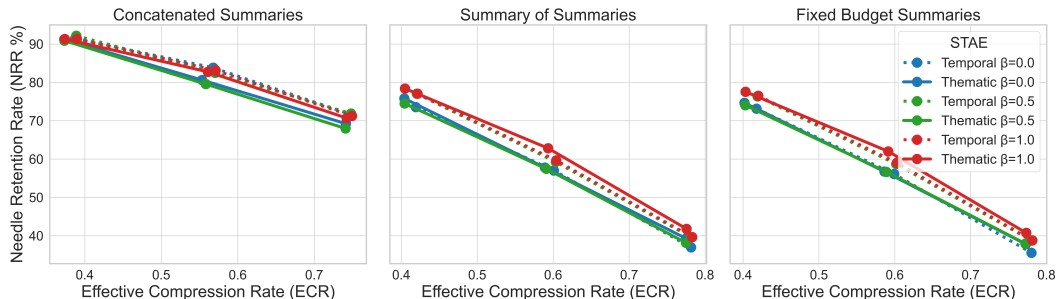

Figure 2: **Experiment 2 (Local STAE): Inverted eviction (evict lowest scores).** NRR (%) vs. *effective* compression (ECR) for temporal vs. thematic grouping under three summarization modes. Temporal (upper-bound baseline) and thematic curves nearly coincide; gaps across $\beta$ are small.

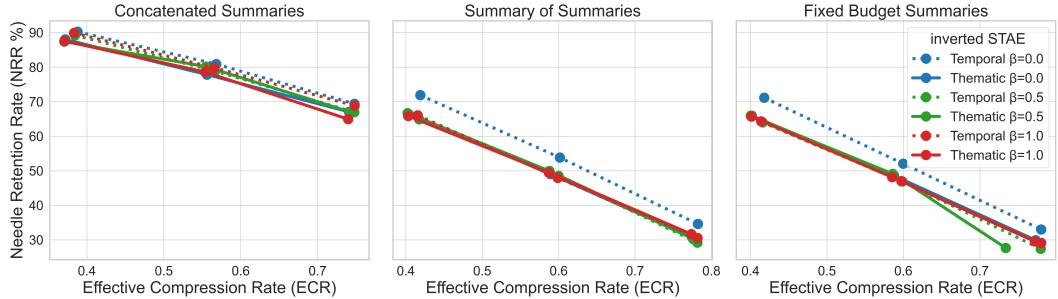

Figure 3: **Experiment 2 (Local STAE): Regular eviction (evict highest scores).** Same setup as Fig. 2 but with regular STAE. Differences relative to inverted are modest locally, with inverted usually slightly better at higher ECR.

that unsupervised semantic grouping is an effective chunking strategy that preserves the integrity of the information even without time cues. This also explains the small gaps across $\beta$: with (roughly) one needle per group, $\beta=0$ and $\beta=1$ perform similarly because local semantic redundancy dominates recency.

**Results.** Across all three summarization modes: (i) temporal and thematic grouping track closely at the same ECR, with temporal only slightly ahead (as expected from the upper-bound argument). (ii) Locally, the gap between inverted and regular is modest and often negligible (though $beta = 1$ achieves largest NRR majority of the time), because redundancy is already localized inside each group. When differences appear (most visible at higher ECR in the fixed-budget setting), inverted has a small but consistent advantage. (iii) For matched ECR, concatenated summaries yields the highest NRR, summary of summaries is lower, and fixed-budget summaries is the most aggressive and thus lowest, reflecting the increasing "information funneling" across these modes.

**Comparison to global STAE.** At comparable ECRs, local STAE dominates global STAE (Sec. 4.1). Group-wise eviction protects distinctive content and prevents globally distinctive but locally rare facts from being compressed away, leading to consistently higher NRR under both temporal and thematic grouping.

**Takeaway.** Localizing STAE to groups, either simple temporal windows or unsupervised embedding-space clusters, delivers strong retention at realistic ECRs. The near-parity between temporal and thematic grouping validates semantic clustering as a practical chunking strategy. The resulting isolated needles per group explains why $\beta=0$ and $\beta=1$ converge.

## 5 DISCUSSION

Across settings, one result is consistent: semantic proximity in embedding space is a strong signal for redundancy, and localizing eviction within groups preserves needles at realistic budgets. Temporal grouping is a strong upper bound on our benchmark (needles are temporally uniform and $k$ is picked so roughly one needle lands per group), and unsupervised thematic clusters closely track the same curves at matched ECR. This near-parity shows that vector-space clustering is an effective chunking strategy even without time cues, reinforcing semantic proximity as a first-class criterion for grouping and compression.

Global eviction must trade off redundancy across unrelated topics, so globally "central" but locally redundant facts can be mistakenly evicted. Group-wise scoring confines redundancy to its topic/-time slice, so summaries stay in context and needles are less likely to mix with off-topic material; empirically, local STAE dominates global STAE at matched ECR across concatenated, summary-of-summaries, and fixed-budget regimes. When groups isolate at most one needle, varying $\beta$ from 0 (FIFO) to 1 (semantic) barely moves NRR-redundancy, not recency, governs. Whereas in global settings $\beta$ matters more because redundancy and topic drift interact over the full history.

Inverting the order clarifies the mechanism: under global scoring, evicting low-scoring (more redundant) units preserves needles better than evicting high-scoring ones; locally, the gap narrows and only remains visible under the most aggressive fixed-budget compression, where inverted eviction yields a small but consistent gain. Because local summarization induces variance in realized budgets, the effective compression rate (ECR) can exceed the nominal target when $k$ is large. Reporting NRR against ECR (not target CR) is therefore essential; the curves reflect the actual budget–accuracy trade-off a practitioner will face.

**Threats to validity and limitations.** Our results rely on off-the-shelf embeddings; domain shift can distort local density estimates. Summary quality (and thus ECR) depends on the LLM and prompt budget and remains stochastic. The 20-needle protocol targets factual retention; higher-order synthesis and causal reasoning are not fully captured. Finally, $k$ (number of groups) controls both redundancy localization and token routing; extremely small or large $k$ can under- or over-fragment topics.

## 6 CONCLUSION

We introduced STAE, a simple, LLM-agnostic policy for redundancy-aware context compression that combines semantic proximity and recency. On a strengthened long-context benchmark with 20 needles per dialogue, we showed: (i) local STAE within groups outperforms global eviction at matched ECR; (ii) temporal chunking provides an upper-bound baseline on our setup, yet thematic clustering by embeddings matches it closely, reinforcing semantic proximity as a principled metric for grouping and/or compressing; and (iii) among eviction orders, removing more redundant (low-score) items generally preserves needles better, with the gap largest in global settings. Together, these findings yield a practical recipe: chunk first, then evict by semantic redundancy, and summarize locally.

**Future work.** We see several directions to broaden and harden these results. Redundancy-aware embedding learning: train encoders with meaning-preserving augmentation (synonym swaps, paraphrase templates) while explicitly minimizing intra-paragraph distances for redundant variants and maximizing distances for contrastive, needle-carrying edits, yielding representations tuned for compression rather than generic similarity. Learned eviction and routing: replace the hand-tuned $\beta$ with a small policy network that predicts keep/evict under a budget, using NRR-at-ECR as reward; explore bandit-style updates or differentiable knapsack surrogates. Radius-based consolidation: complement ranking with a density/radius rule that merges tightly packed messages (keeping a medoid or micro-summary) with radii adapted by robust scale (MAD) or local KDE. Adaptive grouping: learn $k$ and group sizes online (e.g., via BIC or DP-means) to match topic granularity and budget, and study hybrid temporal–thematic partitions. Broader evaluations: go beyond factual needles to multi-hop reasoning and planning, and test cross-domain generalization plus sensitivity to embedding backbones and summarizers. System integration: combine STAE with RAG and memory stores (e.g., KG-augmented caches), measuring end-to-end latency and cost alongside accuracy.

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

## LLM / AI ASSISTANCE DISCLOSURE

In preparing this work, we made use of large language models as auxiliary tools, as follows:

- We used `Claude Sonnet 3.7` to assist in drafting portions of the experimental code (e.g., boilerplate setups, data pipeline scaffolding). All such outputs were reviewed, revised, and validated by the authors before use.
- We used `GPT-4o` and `GPT-5` to polish the manuscript, improve clarity of phrasing, streamline expression, and assist with LaTeX formatting (e.g. reference consistency). All proposed edits were manually vetted and revised by the authors.

The authors retain full responsibility for the content, correctness, and integrity of the final submission, including any errors introduced during LLM-assisted steps.

The LLMs are not credited as authors, since they do not bear responsibility or accountability; authorship credit and accountability remain solely with the human authors.

We provide this disclosure in full transparency in accordance with ICLR 2026's policy requiring that any non-trivial use of LLMs be documented.

