# OpenReview forum: "Semantic Proximity for Redundancy-Aware Context Compression in Large Language Models"
_ICLR.cc/2026/Conference — ICLR 2026 Conference Withdrawn Submission_

### Official Review · Reviewer_K5F3 · 2025-10-30

**Soundness:** 2
**Presentation:** 2
**Contribution:** 2
**Rating:** 2
**Confidence:** 4

**Summary:**

This paper introduces STAE, a novel method for compressing long conversational histories in LLMs by intelligently prioritizing redundant and outdated content based on semantic similarity and recency. The authors develop a robust benchmark with multiple "needles" per dialogue and demonstrate that local compression within semantic or temporal groups significantly outperforms global approaches. The key actionable insight is to first chunk the conversation, then summarize the most redundant parts within each chunk, providing an effective and lightweight solution for managing long contexts.

**Strengths:**

Significant Research Direction:​​ It tackles the critical and practical problem of LLM context window limitation by proposing a lightweight, model-agnostic compression strategy.
​​Clear Writing:​​ The paper is well-structured with a logical flow, from problem definition to experimental validation, making the methodology and contributions easy to follow.

**Weaknesses:**

The STAE approach essentially combines established techniques—embedding, clustering, and linear weighting—into a straightforward scoring mechanism. While practical, it lacks fundamental algorithmic novelty, positioning it more as a pragmatic engineering integration for a specific task rather than a breakthrough in methodology.

The evaluation primarily contrasts STAE against weaker baselines like FIFO, omitting direct comparisons with more advanced memory management methods from the literature, such as MemGPT or SeCom. This absence of rigorous benchmarking makes it difficult to assess STAE's true competitive standing and relative contribution to the field.

The smooth performance curves observed stem from an idealized benchmark with atomized, uniformly distributed "needles." This setup might overstate the method's robustness, as its performance in real-world scenarios requiring complex reasoning or dealing with interdependent information remains unverified and could exhibit non-linear degradation.

The efficacy of STAE is heavily reliant on the underlying embedding model's ability to accurately capture semantic nuances. The lack of ablation studies using different embedding models casts doubt on its generalizability, necessitating additional validation for specific domains and increasing the practical overhead for deployment.

**Questions:**

see the weakness part

---

### Official Review · Reviewer_MBxA · 2025-10-30

**Soundness:** 3
**Presentation:** 2
**Contribution:** 2
**Rating:** 4
**Confidence:** 4

**Summary:**

This paper proposes Semantic–Temporal Aware Eviction (STAE), a lightweight, LLM-agnostic strategy for compressing long conversational histories by prioritizing the removal of semantically redundant and older message pairs. The core idea is to embed user–assistant turn pairs, then score them using a convex combination of (i) semantic redundancy—estimated either via local kernel density or a centroid-proximity proxy—and (ii) normalized recency. High-scoring pairs are evicted and summarized under various budgeting schemes. The authors also introduce a new evaluation benchmark,Needle Density. The key empirical finding is that local STAE—especially with thematic clustering—matches or nearly matches a strong temporal upper bound, while global eviction underperforms. The paper concludes with a practical recommendation: chunk first by time or semantics), then compress redundant content within chunks.

**Strengths:**

- STAE is conceptually simple (a convex interpolation between a semantic redundancy score and normalized age) and can be implemented with off-the-shelf embeddings and lightweight clustering.

- The move to report NRR against realized ECR instead of target CR and to populate dialogues with many needles is a welcome, rigorous step toward measuring real practitioner tradeoffs.

- The experiments convincingly illustrate why local grouping (temporal or thematic) reduces harmful cross-topic evictions and often matches a temporal upper bound, which is an actionable prescription for system designers.

- The paper explores inversion of eviction order, multiple summarization modes (concat / summary-of-summaries / fixed budget), and β sweeps, giving useful insight into which parts of the pipeline matter.

**Weaknesses:**

- The paper’s theoretical motivation in Section 3.2 feels somewhat hand-wavy.  Section 3.2 invokes an ε–δ / manifold-style continuity intuition and latent perturbations induced by human paraphrase operations, but the actual algorithm is quite simple: it measures redundancy via embeddings and evicts centroid-near items. The theoretical framing and the practical method feel loosely connected.

- Although the Needle Density benchmark is an improvement over single-needle NIAH tests, its scope remains limited. The 20 needles are atomic, verifiable facts, uniformly scattered in time—but real agent memory must also preserve reasoning traces, user preferences, or multi-hop dependencies.

- Lack of comparison between the two proposed redundancy estimators: (1) the kernel-density–based score and (2) the centroid-proximity proxy. The paper states the latter is used “for efficiency,” but provides no ablation on either runtime or downstream NRR. This omission weakens the practical guidance the paper aims to offer.

- Finally, the experimental comparisons are largely internal—varying β, grouping strategies, or eviction order. But the paper does not benchmark against context compression such as LLMLingua-2, or even simple RAG retrieval from the raw history.

**Questions:**

1. The pipeline involves multiple stages—embedding all message pairs, clustering, summarizing (and summarizing on summaries), evicted chunks, and finally answering questions. Could you provide a latency or cost breakdown of these components?

2. How robust are your results to the choice of embedding backbone? A brief sensitivity analysis here would strengthen the claim.

---

### Official Review · Reviewer_4XY6 · 2025-11-02

**Soundness:** 3
**Presentation:** 3
**Contribution:** 2
**Rating:** 4
**Confidence:** 3

**Summary:**

The paper tackles long-context LLM limitations by proposing STAE (Semantic–Temporal Aware Eviction), a redundancy-aware context compression policy. It embeds dialogue units (user–assistant pairs), estimates semantic redundancy either via local kernel density or centroid proximity, combines it with normalized recency through a tunable weight β, and evicts/summarizes the most redundant-and-old content. To evaluate this, the authors augment LongMemEval with a harder 20-needle-per-dialogue benchmark and measure Needle Retention Rate (NRR) against Effective Compression Rate (ECR). Results show (i) compressing where semantic redundancy is highest preserves more needles than FIFO; (ii) local STAE (within temporal or semantic clusters) consistently outperforms global eviction at the same ECR; and (iii) semantic clustering is almost as good as temporal windows, making the approach practical for real dialogues without timestamps.

**Strengths:**

**Simple design**

-- STAE is training-free, making it applicable on a wide range of LLMs and tasks. It also avoids extra LLM calls by doing the redundancy test in embedding space.

**Intuitive, clear, actionable objective**

-- “Compress where redundancy is highest” is simple and system-friendly.The proposed method is based the assumption that: (1) dialogue turns that are closer to the overall topic (with low distance to the conversation centroid) is redundant and should be compressed/summarized, while (2) turns that are further to the centroid cannot be represented by other turns and thus should not be compressed.

**Stronger evaluation setup**

-- Extending LongMemEval to 20 evenly distributed needles directly addresses the brittleness of single-needle tests and justifies reporting NRR–ECR curves.

**Reporting ECR (not just target CR)**

-- Acknowledging that summarization drifts from the target budget makes the results more realistic and convincing.

**Weaknesses:**

**Comparison to other context-compression methods**

-- In Sec 4.1 and 4.2, the authors only compared their STAE with an inverted STAE. These experimental results are insightful, but I believe it is still necessary to at least compare STAE with some existing state-of-the-art context-compression methods.

**Potential overloading the notation $k$**

-- In Sec 3.4, $k$ is the number of nearest neighbors of m within group G considered when calculating kernel density; while in Sec 3.5, $k$ is the number of clusters/groups. From my current understanding of this work, these are two different quantities.

**Confused between two versions of the semantic redundancy**

-- In Sec 3.4, the authors defined an embedding-space-density-based semantic redundancy (more precise) and a centroid-based proxy (faster). However, it is not clear which one is used in Sec 4 experiments. And the abstract is written with respect to the centroid-based proxy only, leaving me wondering  “where exactly is the centroid used in computing the semantic redundancy?” when I was reading Sec 3.5 for the first time.

**Questions:**

-- Is it possible to dynamically select k (number of clusters) by detecting topic shifts (hence deciding how many centroids to use) using the LLM itself, without relying on silhouette analysis?

-- Summarization noise is underexplored: stochastic LLM summaries can drop needles; the paper reports the drift (via ECR) but doesn’t quantify how often summaries themselves are the bottleneck.

-- Can you implement STAE in a compress + retrieve pipeline that incorporate RAG in handling evicted turns?

-- Could $\beta$ be learned or scheduled (e.g., start temporal, become semantic as topics stabilize) to reduce the need for manual tuning?

---

### Official Review · Reviewer_r9pE · 2025-11-03

**Soundness:** 3
**Presentation:** 3
**Contribution:** 3
**Rating:** 2
**Confidence:** 3

**Summary:**

This paper tackles the critical problem of limited context windows in Large Language Models (LLMs) by proposing a context compression strategy for conversational histories. The authors argue that naive compression methods, such as FIFO, discard crucial facts.

To address this, the authors propose **STAE (Semantic-Temporal Aware Eviction)**, a lightweight eviction policy. The strategy involves pairing and embedding (user, assistant) dialogue turns, then ranking these pairs by a score $S(m)$, which is a convex combination of **semantic redundancy $R(m)$** (via distance to a centroid or kernel density estimation) and **recency $T(m)$**. The core idea is that pairs that are semantically redundant and temporally old should be prioritized for compression (summarization).

To evaluate information retention under compression, the authors also contribute a **new benchmark**. They augment LongMemEval to create a **"Needle Density Benchmark" with 20 needles per dialogue**, which is more robust than standard single-needle tests (like NAIH). The evaluation metric is the **Needle Retention Rate (NRR)** versus the **Effective Compression Rate (ECR)**.

Experiments compare "global" application of STAE (over the whole dialogue) versus "local" application (within temporal chunks or semantic clusters).

The main findings include:
1.  **Local > Global:** Applying the STAE policy within local groups (either temporal or thematic) is more effective at preserving needles than applying it globally.
2.  **Semantic Clustering is an Effective Chunking Strategy:** Thematic clustering via k-means on embeddings (an unsupervised method) performs nearly as well at preserving needles as "temporal chunking". This shows semantic proximity is a practical and effective criterion for chunking.
3.  **Efficacy of Redundancy-Awareness:** Despite severe confusion in the presentation, the experiments appear to show that evicting (compressing) the most semantically redundant items—rather than the most unique ones—is the correct strategy for preserving key information.

**Strengths:**

1.  **Important and Practical Problem:** How to compress an LLM's context without losing key facts is a core, urgent challenge for deploying long-running conversational agents.
2.  **Robust Benchmark:** The "20-needle density benchmark" is a strong contribution. It is more difficult and realistic than single-needle tests and properly evaluates the information-preserving capacity of a compression strategy. The explicit focus on NRR vs. ECR is good practice.
3.  **Lightweight and Efficient Method:** The STAE policy itself (computing centroids and distances in embedding space) is lightweight and **requires no extra LLM calls** for scoring, making it computationally attractive compared to other methods that might rely on an LLM for "meta-reflection".
4.  **Clear Experimental Takeaways (Locality):** The "local > global" and "thematic clustering ~= temporal chunking" findings are clear, actionable insights. They provide a practical recipe for practitioners on how to build a context compressor: chunk first, then compress.

**Weaknesses:**

1.  **Critical Confusion in Core Result (Fatal Flaw):** The paper suffers from a fundamental, confusing contradiction in its presentation of the core global STAE results (Section 4.1).
    * **The Math:** Section 3.4 defines the score $S(m)$ as a combination of redundancy $R(m)$ (high=redundant) and age $T(m)$ (high=old). Therefore, a *high* $S(m)$ score means "old and redundant" and is "most evictable".
    * **The Plot:** Figure 1, via its caption and legend, shows that "Lowest" (solid lines, i.e., evicting the *lowest*-scoring items) *consistently outperforms* "Highest" (dashed lines, i.e., evicting the *highest*-scoring items).
    * **The Text:** The interpretation text in Section 4.1 (“evicting low-score (semantically redundant) pairs preserves...”) suggests that "low-score" means "redundant".
    * **This is a total contradiction.** The math ("high-score = redundant") conflicts with the text ("low-score = redundant") and the plot (which shows "evicting low-score" is best). This makes it impossible for the reader to determine what policy was *actually* run or to trust the conclusion of Section 4.1. This contradiction must be resolved before the paper can be accepted.
2.  **Benchmark Design Limitations:** The needles are distributed *uniformly* in time. This makes the "temporal chunking" strategy a powerful, but possibly *artificial*, upper bound. In real-world dialogues, needles (key facts) are likely clustered by topic, which may not align with contiguous temporal chunks.
3.  **Dependence on Method Components:** The results are dependent on the specific embedding model and summarization model. The summarization step, in particular, is critical to the NRR, yet its quality is not analyzed.

**Questions:**

1.  **(CRITICAL)** Please clarify the fundamental contradiction in your paper:
    * (a) Is the mathematical definition in Section 3.4 correct? i.e., a high $S(m)$ score means "redundant and old"?
    * (b) If (a) is true, why does Figure 1 show that evicting the "Lowest" score (i.e., new, unique items) gives the *best* performance? This seems counter-intuitive.
    * (c) If (a) is false, what is the correct definition of $S(m)$?
    * (d) Why does the interpretation text in Section 4.1 ("low-score (semantically redundant)") contradict the mathematical definition in Section 3.4 (high-score = redundant)?
2.  The 20-needle benchmark distributes needles uniformly in time. How do you think your results (especially temporal vs. thematic clustering) would change if needles were distributed non-uniformly, e.g., clustered by *topic* which might span non-contiguous time blocks?
3.  You use k-means for thematic clustering. How sensitive are the results to the choice of $k$ (number of clusters)? You mention $k=25$ was "selected via silhouette analysis," but this seems like a fixed value. How does the choice of $k$ impact the ECR-NRR trade-off?

---

### Note · Authors · 2025-11-28

**Comment:**

After careful consideration, we have decided to withdraw this submission at this stage. We intend to substantially expand the experimental section and broader empirical analysis, so that a future version can more fully showcase the strengths, limitations, and practical impact of our method. We also plan to incorporate the reviewers’ suggestions and clarifications into a more polished and comprehensive revision. We are grateful to the program chairs, senior area chairs, area chairs, and reviewers for their valuable feedback, time, and consideration.

**Withdrawal Confirmation:**

I have read and agree with the venue's withdrawal policy on behalf of myself and my co-authors.